# Modeling of the Achilles Subtendons and Their Interactions in a Framework of the Absolute Nodal Coordinate Formulation

**DOI:** 10.3390/ma15248906

**Published:** 2022-12-13

**Authors:** Leonid P. Obrezkov, Taija Finni, Marko K. Matikainen

**Affiliations:** 1Faculty of Sport and Health Sciences, University of Jyväskylä, 40014 Jyväskylä, Finland; 2Mechanical Engineering, LUT University, 53850 Lappeenranta, Finland

**Keywords:** biomechanics, Achilles tendon, beam-to-beam contact, arbitrary cross-section, ANCF, elasticity

## Abstract

Experimental results have revealed the sophisticated Achilles tendon (AT) structure, including its material properties and complex geometry. The latter incorporates a twisted design and composite construction consisting of three subtendons. Each of them has a nonstandard cross-section. All these factors make the AT deformation analysis computationally demanding. Generally, 3D finite solid elements are used to develop models for AT because they can discretize almost any shape, providing reliable results. However, they also require dense discretization in all three dimensions, leading to a high computational cost. One way to reduce degrees of freedom is the utilization of finite beam elements, requiring only line discretization over the length of subtendons. However, using the material models known from continuum mechanics is challenging because these elements do not usually have 3D elasticity in their descriptions. Furthermore, the contact is defined at the beam axis instead of using a more general surface-to-surface formulation. This work studies the continuum beam elements based on the absolute nodal coordinate formulation (ANCF) for AT modeling. ANCF beam elements require discretization only in one direction, making the model less computationally expensive. Recent work demonstrates that these elements can describe various cross-sections and materials models, thus allowing the approximation of AT complexity. In this study, the tendon model is reproduced by the ANCF continuum beam elements using the isotropic incompressible model to present material features.

## 1. Introduction

The Achilles tendon (AT) is the strongest tendon in the body and serves an important function during locomotion. It can reach loads up to four times the body weight while walking and approximately 10 times while running, with the upper border as much as 9 kN [1]. At the same time, it is vulnerable to traumatic injuries due to chronic or acute overloading [2], with the determinants of good recovery not well understood. The AT possesses a complex structure whereby three subtendons, having subject-specific cross-sections, arise from the soleus and the lateral and medial heads of the gastrocnemius muscles. The three heads twist around each other, counterclockwise in the right AT and clockwise in the left. With the complex structure comes functional consequences. Studies have revealed nonuniform displacements within healthy AT [3], whereas the displacement can be more uniform in an injured tendon [4]. Furthermore, loading from the three different muscles causes nonhomogeneous longitudinal strains, compression, and transverse strains in the AT [5]. Longitudinal and transverse strains have also been reported in human studies [6,7]. These nonhomogeneous strains are likely linked to the architectural structure of the tendon [8]. The AT can endure the large forces transmitted axially, and they are most often studied, whereas less attention is placed on shear forces. Because of its significant role in the human musculoskeletal system, a better understanding of AT can provide valuable information for diagnosis and treatment.

What is the significance of the complex geometry and twist of AT? Previous research has shown that region-specific susceptibility to strain injury changes with the amount of tendon twist [9] and that the changes in AT stress are more sensitive to volumetric tendon shape rather than material properties [10]. The appropriate model can help study stress and strain distributions within the AT and improve understanding of tendon function in health, disease, and rehabilitation.

One of the most popular methods in modern biomechanical research for studying tendons is the finite element method (FEM). Using finite solid elements in a framework of the nonlinear FEM helps to comprehend the tendon’s complex geometry and materiality. However, AT modeling with the solid finite elements leads to a significant number of degrees of freedom (DOFs) [11,12,13], which results in a long computation time to obtain a solution. Solid elements can approximate almost any shape and provide reliable results, but require dense discretization in all three dimensions. Therefore, other types of finite elements are necessary to decrease the computational cost.

This research introduces a new approach to the deformation analysis of the Achilles subtendons and their interactions. The main idea is to use one-dimensional finite element discretization over the subtendon’s length (in the longitudinal direction) to decrease computational costs. It can be achieved by considering the tendon as a beam-like structure using ANCF-based continuum beam elements with specific descriptions for geometrically complicated deformable cross-sections. This approach leads to the finite element discretization over the subtendon’s length and makes it possible to consider material laws based on the continuum mechanics. Recent studies [14,15,16,17] show that this transition is possible without significant losses in the quality of the results within the ANCF framework. For example, the work [16] considers the deformations of the beam-like structures described with the ANCF continuum beam elements and from the various soft material models. In [17], the approximation of the rat Achilles tendon experiment with the ANCF elements is given and verified against experimental results. Ref. [15] provides the approximation way for arbitrary cross-sections, which is suitable for the continuum-based ANCF beams. For example, the provided technique approximates one of the Achilles subtendons. In the case of multibeam construction, the question of mutual interaction between beams arises, i.e., the so-called contact problem. In the work [14], the methods for solving contact problems between beams with arbitrary cross-sections are presented.

In this study, we explored the method given in [15] and modeled the whole AT with the ANCF continuum beam elements. The subtendons’ cross-sections are obtained with the integration scheme proposed in [15]. Then, the obtained beams are pretwisted, one around the other. In this study, the neo-Hookean material model describes the soft tissue response [10,18]. There are also possibilities to use others’ material models in a way given in [16]. However, Annaidh et al. [19] demonstrated that using anisotropic material models within FEM can have inconsistencies. The possible contact between subtendons can be described via the surface-to-surface procedure, thus, taking into account the complicated border interactions between two bodies.

## 2. ANCF Beam Element

This section provides the geometrical setup for the continuum-based ANCF beam element. The idea behind this element type is to use the slope vectors for defining the cross-section orientation and deformation. The advantages of these finite elements are discussed in [20,21,22].

There are various types of ANCF elements, and one can divide them into several groups and subgroups; for more details, the reader is referred to Nachbagauer et al. [23], Obrezkov et al. [24], Patel and Shabana [25]. Here, the higher-order three-nodded element with the second-order interpolation in longitudinal and thickness directions denoted 3363 is used. It does not require any modifications to demonstrate good performance even for complicated loading cases [26] and allows the use of all material laws based on the 3D elasticity.

### 2.1. Kinematics of the ANCF Continuum Beam Elements

Let $r=r\left(x,y,z\right)\in R^{3}$ be the position vector field of any particle in the current configuration. The position in the initial configuration is denoted as $\bar{r}$ [15], see Figure 1. The connection between the two vectors is
(1)$$r=\bar{r}+u_{h},$$
where $u_{h}$ is a displacement vector. Hence, the body motion is
(2)$$r\left(x,y,z,t\right)=N_{m}\left(x,y,z\right)q\left(t\right),$$
where $N_{m}$ is a shape function matrix and q is a vector of nodal coordinates. q contains the position of the nodes as well as their derivatives. Therefore, we accept the following notation for ith node:$$r_{,x}^{i}=\frac{\partial r^{i}}{\partial x},r_{,y}^{i}=\frac{\partial r^{i}}{\partial y},r_{,z}^{i}=\frac{\partial r^{i}}{\partial z},$$
$$r_{,yy}^{i}=\frac{\partial^{2}r^{i}}{\partial y^{2}},r_{,yz}^{i}=\frac{\partial^{2}r^{i}}{\partial yz},r_{,zz}^{i}=\frac{\partial^{2}r^{i}}{\partial z^{2}}.$$

As mentioned above, in this work, we consider 3363 beam elements [26]. The vectors of nodal coordinates related to this element are presented as follows:(3)$$q^{i}=\left[r^{i},r_{y}^{i},r_{z}^{i},r_{yy}^{i},r_{zz}^{i},r_{yz}^{i}\right].$$

Accordingly, the vector of displacements $u_{h}$ has the form
(4)$$u_{h}\left(x,y,z,t\right)=N_{m}\left(x,y,z\right)u\left(t\right),$$
where u is a vector of nodal displacements. The element is isoparametric. Here, we introduce a new local coordinate system $\xi =\left\{\xi ,\eta ,\zeta \right\}$ with the range for the local coordinates $\left[-1,1\right]$, where $\xi =\frac{2x}{l_{x}},\eta =\frac{2y}{l_{y}},\zeta =\frac{2z}{l_{z}}$. Here, $l_{x},l_{y}$ and $l_{z}$ are the physical dimensions of the element. The substitutions are made to deal with the Gaussian integration procedure [15]. Now, we have
(5)$$\begin{array}{c} r\left(\xi ,\eta ,\zeta ,t\right)=N_{m}\left(\xi ,\eta ,\zeta \right)q\left(t\right),\\ u_{h}\left(\xi ,\eta ,\zeta ,t\right)=N_{m}\left(\xi ,\eta ,\zeta \right)u\left(t\right). \end{array}$$

Then, the form of the shape function matrix is
(6)$$N_{m}\left(\xi ,\eta ,\zeta \right)=\left[N_{1}I\;N_{2}I\;N_{3}I...N_{18}I\right],$$
where I is a 3×3 identity matrix and components of $N_{m}$ are
$$\begin{array}{ccc} N_{1}=\frac{1}{2}\xi \left(\xi -1\right) & N_{2}=\frac{1}{4}l_{y}\xi \eta \left(\xi -1\right) & N_{3}=\frac{1}{4}l_{z}\xi \zeta \left(\xi -1\right)\\ N_{4}=\frac{1}{8}l_{z}l_{y}\xi \eta \zeta \left(\xi -1\right) & N_{5}=\frac{1}{16}l_{y}^{2}\xi \eta^{2}\left(\xi -1\right) & N_{6}=\frac{1}{16}l_{z}^{2}\xi \zeta^{2}\left(\xi -1\right)\\ N_{7}=1-\xi^{2} & N_{8}=\frac{1}{2}l_{y}\eta \left(1-\xi^{2}\right) & N_{9}=\frac{1}{2}l_{z}\zeta \left(1-\xi^{2}\right)\\ N_{10}=\frac{1}{4}l_{z}l_{y}\eta \zeta \left(1-\xi^{2}\right) & N_{11}=\frac{1}{8}l_{y}^{2}\eta^{2}\left(1-\xi^{2}\right) & N_{12}=\frac{1}{8}l_{z}^{2}\zeta^{2}\left(1-\xi^{2}\right)\\ N_{13}=\frac{1}{2}\xi \left(\xi +1\right) & N_{14}=\frac{1}{4}l_{y}\xi \eta \left(\xi +1\right) & N_{15}=\frac{1}{4}l_{z}\xi \zeta \left(\xi +1\right)\\ N_{16}=\frac{1}{8}l_{z}l_{y}\xi \eta \zeta \left(\xi +1\right) & N_{17}=\frac{1}{16}l_{y}^{2}\xi \eta^{2}\left(\xi +1\right) & N_{18}=\frac{1}{16}l_{z}^{2}\xi \zeta^{2}\left(\xi +1\right). \end{array}$$

For further investigation, it is necessary to define the deformation gradient F. From (Equation 1) and (Equation 2), it can be written as
(7)$$F=\frac{\partial r}{\partial \bar{r}}=\frac{\partial r}{\partial \xi }{\left(\frac{\partial \bar{r}}{\partial \xi }\right)}^{-1}=I+\frac{\partial u_{h}}{\partial \xi }{\left(\frac{\partial \bar{r}}{\partial \xi }\right)}^{-1}.$$

The determinant of F defines the volume ratio of the element, we assume
(8)J=detF>0.

### 2.2. Cross-Section Geometry Description

The standard Gaussian quadrature formula for the integration of any function $f\left(x,y\right)$ in the general form can be written as follows,
(9)$$\int_{\Omega }f\left(x,y\right)d\Omega =\sum_{i=1}^{n}\sum_{j=1}^{n}f\left(x_{i},y_{j}\right)w_{i}w_{j},$$
where 2n−1 is the polynomial exactness degree of function *f* over one of the axis lines, and *w* is the weight of the point. For simple cross-sections (circular, etc.), we send our readers to [27], where weights and points in a binormalized coordinate system can be found. Below we present the method for more complicated domains, which can also be found in [15].

Let us consider a closed domain Ω, which has a piecewise border ∂Ω with points $V_{i}$ on it:(10)$$V_{i}=\left(\alpha_{i},\beta_{i}\right),i=1,..,\varphi ,$$
$$\partial \Omega =\left[V_{1},V_{2}\right]\cup \left[V_{2},V_{3}\right]\cup ...\cup \left[V_{\varphi },V_{1}\right].$$

The lines $\left[V_{i},V_{i+1}\right]$ also have several additional “control” points, such as $P_{i1}=V_{i},P_{i2},...,P_{im_{i}}=V_{i+1}$, or in the binormalized coordinates as $P_{i1}^{\xi }=V_{i}^{\xi },...,P_{im_{i}}^{\xi }=V_{i+1}^{\xi }$. Subsequently, the “cumulative chordal” formula parametrization is recalled:$$\left[\alpha_{ij}^{\xi },\beta_{ij}^{\xi }\right]=\left[0,\sum_{j=1}^{m_{i}-1}\Delta t_{ij}\right],|\Delta t_{ij}|=|P_{ij+1}^{\xi }-P_{ij}^{\xi }|,j=1,...,m_{i}-1.$$

Then, each line $\left[V_{i}^{\xi },V_{i+1}^{\xi }\right]$ is tracked by a spline curve $S_{i}\left(t\right)=\left(S_{i1}\left(t\right),S_{i2}\left(t\right)\right)$ degree of $p_{i}$, where $p_{i}\leq m_{i}-1$, see Figure 2.

Then the cubature formula with the 2n−1 polynomial exactness degree over the Ω domain has the form
(11)$$I_{2n-1}=\sum_{\lambda \in \Lambda_{2n-1}}f\left(\eta_{\lambda },\zeta_{\lambda }\right)w_{\lambda },$$
where
$$\Lambda_{2n-1}=\{\lambda =\left(i,j,k,h\right):1⩽i⩽\varphi ,1⩽j⩽m_{i}-1,$$
$$1⩽k⩽n_{i},1⩽h⩽n\},$$
and $w_{\lambda }$, $\eta_{\lambda }$ and $\zeta_{\lambda }$ are:$$\eta_{\lambda }=\frac{S_{i1}\left(q_{ijk}\right)-\Xi }{2}\tau_{h}^{n}+\frac{S_{i1}\left(q_{ijk}\right)+\Xi }{2},$$
$$\zeta_{\lambda }=S_{i2}\left(q_{ijk}\right),$$
$$w_{\lambda }=\frac{\Delta t_{ij}}{4}w_{k}^{n_{i}}w_{h}^{n}\left(S_{i1}\left(q_{ijk}\right)-\Xi \right)\frac{dS_{i2}\left(t\right)}{dt}{|}_{t=q_{ijk}},$$
$$q_{ijk}=\frac{\Delta t_{ij}}{2}\tau_{k}^{n_{i}}+\frac{t_{ij+1}+t_{ij}}{2},\Delta t_{ij}=t_{ij+1}-t_{ij},$$
$$n_{i}=\left\{\begin{array}{c} np_{i}+p_{i}/2,p_{i}\mathrm{is}\;\mathrm{even},\\ np_{i}+\left(p_{i}+1\right)/2,p_{i}\mathrm{is}\;\mathrm{odd}. \end{array}\right.$$

Thus, only $\tau_{k}^{n_{i}}$, $w_{k}^{n_{i}}$ and Ξ need to be defined. Ξ is an arbitrary straight line
$$\Omega \subseteq R^{2}=\left[a,b\right]\times \left[c,d\right],\Xi \left(\eta \right)\in \left[a,b\right],\eta \in \left[c,d\right].$$

The choice of Ξ does not have any influence. However, it is necessary to obtain the nodes and weights. $\tau_{k}^{n_{i}}$, $w_{k}^{n_{i}}$ are the nodes and weights, respectively, of the Gauss–Legendre quadrature formula of the exactness degree $2n_{i}-1$ on [−1,1].

## 3. Equilibrium Equation

Our task involves many subroutines, each of them contributing to the energy balance and equilibrium of the whole system. The common approach for calculating is to use the variational formulation. The variations can be grouped as inertia, external, contact, and internal:(12)$$\delta \Pi_{ext}-\delta \Pi_{int}+\delta \Pi_{inert}-\delta \Pi_{con}=0.$$

$\delta \Pi_{inert}$ can be written as
(13)$$\delta \Pi_{inert}={\ddot{q}}^{T}\int_{V}\rho N^{T}NdV\;\cdot \;\delta q,$$
where ρ is the mass density, and *V* is the volume of the element in the reference configuration. The mass matrix is $M=\int_{V}\rho N^{T}NdV$. In the case of the static problem, which is the concern of this work, $\delta \Pi_{inert}=0$. The variation of $\Pi_{int}$ with respect to the nodal coordinates is [16]
(14)$$\delta \Pi_{int}=\int_{V}S:\delta EdV=\int_{V}S:\frac{\partial E}{\partial q}dV\;\cdot \;\delta q.$$

S is the second Piola–Kirchhoff stress tensor, and its form depends on the material model, which will be presented in Section 4. E is the Green–Lagrange strain tensor
(15)$$E=\frac{1}{2}\left(F^{T}\;\cdot \;F-I\right).$$

The last is the variation of the contact force work $\delta \Pi_{con}$, which will be explained.

## 4. Approximation of the Tendon Tissue

The elastic properties of the Achilles tendon tissue can be presented in different ways. One can find examples in [10,12,13,28,29], where the Helmholtz free energy function Ψ describes elastic features for such material models. In the isotropic case, Ψ depends only on the right Cauchy–Green tensor $C=F^{T}\;\cdot \;F$, therefore, Ψ=Ψ(C). In the case of anisotropy, the additional structural tensor A can be added to define the preferable deformation direction Ψ=Ψ(C,A). Models describing AT are usually incompressible. The common approach to deal with it is to split the deformation gradient F into dilational (volumetric) and distortion (isochoric) parts. Here again, we want to send our readers to the work [19], where the authors point out the possible problems associated with the decomposition of the anisotropic material models. We have
(16)$$F=J^{\frac{1}{3}}\bar{F},J=\det\;F>0.$$

This leads to the follow representation of the right Cauchy–Green tensor:(17)$$\bar{C}={\bar{F}}^{T}\;\cdot \;\bar{F}.$$

Thus, after the decomposition, we have
(18)$$\Psi =\Psi_{vol}\left(J\right)+\Psi_{iso}\left(\bar{C},\right),$$
where $\Psi_{vol}\left(J\right)=k{\left(J-1\right)}^{2}$, *k* is a penalty coefficient to guarantee the incompressibility. The part $\Psi_{iso}$ might be reformulated in the terms of the Cauchy–Green deformation tensor invariants,
(19)$$\Psi_{iso}=\Psi_{iso}\left(\bar{I_{1}},\bar{I_{2}}\right),$$
where $\bar{I_{1}}$ and $\bar{I_{2}}$ have the forms
(20)$$\begin{array}{c} {\bar{I}}_{1}=\mathrm{tr}\bar{C},\\ {\bar{I}}_{2}=\frac{1}{2}\left(\mathrm{tr}\;{\bar{C}}^{2}+{\mathrm{tr}}^{2}\bar{C}\right). \end{array}$$

The second Piola–Kirchhoff stress from (Equation 14) is formulated as follows:(21)$$S=2\frac{\partial \Psi }{\partial C}=2\frac{\partial \Psi }{\partial \bar{C}}:\frac{\partial \bar{C}}{\partial C}.$$

Using (Equation 18), it can be expressed as
(22)$$S=2\frac{\partial \bar{\Psi }}{\partial \bar{C}}\frac{\partial \bar{C}}{\partial C}+2\frac{\partial \Psi_{vol}}{\partial J}\frac{\partial J}{\partial C}=2\left(\sum_{k}\frac{\partial \bar{\Psi }}{\partial {\bar{I}}_{k}}\frac{\partial {\bar{I}}_{k}}{\partial \bar{C}}\right)\frac{\partial \bar{C}}{\partial C}+\frac{\partial \Psi_{vol}}{\partial J}JC^{-1},$$
$$\frac{\partial J}{\partial C}=\frac{1}{2}JC^{-1}.$$

The corresponding volumetric part has the form
(23)$$S_{vol}=d\left(J-1\right)JC^{-1}.$$

In this study, we consider one type of material model: the neo-Hookean model. The isochoric part of the neo-Hookean model is
(24)$$\bar{\Psi }=c_{10}\left(\bar{I_{1}}-3\right),$$
with the expression for the second Piola–Kirchhoff stress tensor:(25)$$\bar{S}=2c_{10}J^{-\frac{2}{3}}\left[I-\frac{1}{3}{\bar{I}}_{1}{\bar{C}}^{-1}\right].$$

## 5. Contact Formulation

Working with an assembled structure consisting of two or more bodies, the question of interaction between the substructures appears. That problem requires the solution of a contact task. In this study, we are concerned with the description of the bodies of nonstandard forms, such as only surface-to-surface contact formulation, which can describe this contact [14].

Let us describe the task of two contacting beams (denoted as *A* and *B*) in the terms of the distances between the two closest position vector fields $r^{A}$ and $r^{B}$. Then, assuming that along the contact surface there is no penetration, the minimum distance problem in the most general case can be formulated as follows:(26)$$d=r^{A}-r^{B}.$$

The nonpenetration condition is defined via the so-called gap function, which in this work is given as follows,
(27)$$\begin{array}{cc} g(\xi^{A},\xi^{B},\eta^{A},\eta^{B},\zeta^{A},\zeta^{B}) & =r^{A}{\left(\xi^{A}\right)}_{\eta ,\zeta =0}-r^{B}{\left(\xi_{c}^{B}\right)}_{\eta ,\zeta =0}\\ & -(r^{A}\left(\xi^{A},\eta^{A},\zeta^{A}\right)-r^{A}{\left(\xi^{A}\right)}_{\eta ,\zeta =0}\\ & +r_{c}^{B}\left(\xi_{c}^{B},\eta_{c}^{B},\zeta_{c}^{B}\right)-r^{B}{\left(\xi_{c}^{B}\right)}_{\eta ,\zeta =0}), \end{array}$$
where $g(\xi^{A},\xi^{B},\eta^{A},\eta^{B},\zeta^{A},\zeta^{B})\geq 0$. The subscript ${}_{c}$ denotes the orthogonal projection of the point on beam *A* on the beam *B* obtained from (Equation 27), $r^{B}{\left(\xi_{c}^{B}\right)}_{\eta ,\zeta =0}$ is the point projection $r_{c}^{B}\left(\xi_{c}^{B},\eta_{c}^{B},\zeta_{c}^{B}\right)$ on the beam centerline. In the model, we assume that two bodies are closely placed to each other, and only sliding is allowed. Therefore, the nonpenetration condition is g=0. Then the variation reads as follows,
(28)$$\delta \Pi_{con}=p_{n}\int_{\Omega_{c}}g\delta g\;d\Omega ,$$
where $\Omega_{c}$ is the contacting surface between *A* and *B* beams, and $p_{n}$ is the penalty parameter. The weak form of contact energy (Equation 28) presented in Section 3 can be expressed in the discrete form as follows,
(29)$$\begin{array}{cc} \delta \Pi_{\mathrm{con}} & =-\delta u_{A}^{T}p_{n}\sum_{i=1}^{n^{i}}\sum_{j=1}^{n^{j}}\sum_{k=1}^{n^{k}}g\left(\xi_{i}^{A},\eta_{j}^{A},\zeta_{k}^{A}\right)N_{A}^{T}n_{ijk}w_{i}w_{j}w_{k}\\ & +\delta u_{B}^{T}p_{n}\sum_{i=1}^{n^{i}}\sum_{j=1}^{n^{j}}\sum_{k=1}^{n^{k}}g\left(\xi_{c}^{B},\eta_{c}^{B},\zeta_{c}^{B}\right)N_{B}^{T}n_{ijk}w_{i}w_{j}w_{k}, \end{array}$$
where
$$n_{ijk}=n\left(\xi_{c}^{B}\left(\xi_{i}^{A}\right),\eta_{c}^{B}\left(\xi_{i}^{A},\eta_{j}^{A},\zeta_{k}^{A}\right),\eta_{c}^{B}\left(\xi_{i}^{A},\eta_{j}^{A},\zeta_{k}^{A}\right)\right),$$
$$N_{A}^{T}=N^{T}\left(\xi_{i}^{A},\eta_{j}^{A},\zeta_{k}^{A}\right)$$
$$N_{B}^{T}=N^{T}\left(\xi_{c}^{B}\left(\xi_{i}^{A}\right),\eta_{c}^{B}\left(\xi_{i}^{A},\eta_{j}^{A},\zeta_{k}^{A}\right),\zeta_{c}^{B}\left(\xi_{i}^{A},\eta_{j}^{A},\zeta_{k}^{A}\right)\right).$$

In (Equation 29), $n^{i}$ is the amount of Gauss points in the *A* beam, along the ξ direction, $w_{i}$ are their corresponding weight, $\eta_{k}$ and $\zeta_{k}$ are the Gauss points coordinates along the η and ζ directions parameters. $\xi_{c}^{B}$, $\eta_{c}^{B}$ and $\zeta_{c}^{B}$ are the parameters of the closest projected point $r(\xi_{j}^{A}$, $\eta_{k}^{A}$, $\zeta_{k}^{A})$ on *B*, n is a normal vector from the *B* to *A* beam elements’ surfaces.

## 6. Numerical Examples

Previous studies found that the Achilles tendon consists of three subtendons with each having a complicated cross-section shape [30,31]. Additionally, there are three common types of AT with varying subtendon regions and torsion [30]. In this work, we consider the AT of Type III due to its relatively simple cross-section form. We extracted the geometrical description of subtendons from [30]. Although the exact geometrical data are not presented, we use CAD software to obtain the positions of the points, as in [15]. Here, we also considered the pretwist of the tendon about the centroidal axis (line, where all three subtendons are connected) from $0^{\circ }$ at x=0 to ψ degrees at x=L. The centroidal axis of the beam remains straight. See Figure 3. The representations of the Gauss points for all three subtendons are given in Figure 4a–c.

The length of the tendon was set at L=0.07 m [29]. The geometrical results based on the approximation are 16.31
${\mathrm{mm}}^{2}$ for the soleus subtendon, 15.98
${\mathrm{mm}}^{2}$ for the medial and 19.57
${\mathrm{mm}}^{2}$ for lateral subtendons, with total area equaling to 51.86
${\mathrm{mm}}^{2}$. That slightly exceeds the average female tendon cross-section 51.2
${\mathrm{mm}}^{2}$ and is smaller than the average male cross-section 62.1
${\mathrm{mm}}^{2}$ [29].

We used the neo-Hookean material model with three shear modulus equal to $c_{10}=103.1$ MPa for soleus, $c_{10}=143.2$ MPa, and $c_{10}=226.7$ MPa for medial and lateral subtendons, respectively [29]. We considered three different pretwisted designs: ψ=0, ψ=15, and ψ=45. The choice is based on the work [28], where the optimal value of twisted is found between 15 and 45 degrees. Then, the soleus subtendon was subjected to forces along the longest direction and applied at the last node, the maximum applied tensile load is 400 N. The applied force exceeds four times the loading conditions given in [15,29], allowing the demonstration of the nonlinear deformations, about 10% of the initial length. On the other edge, r=0 from (Equation 3) is fixed at the first node, and this condition forbids the displacement, but allows the cross-sectional contraction.

The results presented in Table 1 are consistent with the ones given in [15], where the pretwisted subtendons show higher elongations under the same load in comparison to straight subtendons. The elongations for other subtendons are near zero, which indicates that there is sliding between the subtendons as in Section 5 holds. Table 2 presents the converge tests, wherein the elongation results for a number of mesh refinements for the straight and pretwisted soleus subtendon of Type III from the neo-Hookean material model subjected to N=400 N tensile force are given.

The deformed shapes for straight pretwisted ψ=15 the tendons are given in Figure 5a, where the shapes are discretized by four ANCF-based continuum beam elements at each subtendon.

## 7. Conclusions

This work uses the continuum-based ANCF beam element to describe the human Achilles tendon’s deformation due to elongation. In the study, the AT is presented as a combination of three substructures, pretwisted and sliding one around the others. The contact between them is described with the segment-to-segment algorithm. The Gauss–Green cubature integration formula captures the sophisticated cross-section form of each subtendon. The neo-Hookean isotropic material model describes the pure elastic response. The results show that the model is feasible, but more careful verification is necessary. That can include models built with conventional 3D solid elements and the comparison with experimental data.

Additionally, the work possesses certain limitations. For example, the cross-sectional area is taken to be the same for all subtendons along their longitudinal axes. That is a substantial simplification, but there is no available geometrical data to approximate such variation.

## Figures and Tables

**Figure 1 materials-15-08906-f001:**
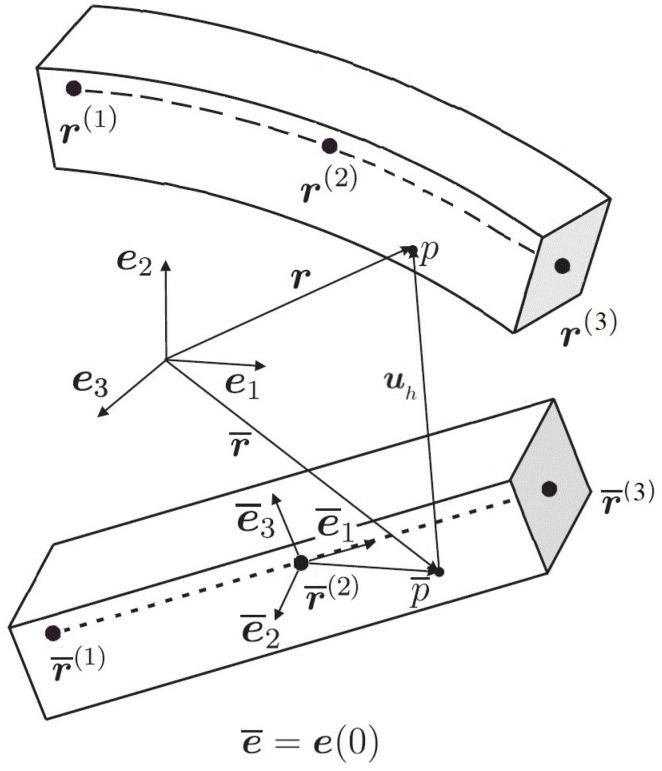
Illustration of a three-nodded beam element with an arbitrary particle *p* in current and $\bar{p}$ in reference configurations. The three nodes are denoted by $r^{\left(i\right)}$ and ${\bar{r}}^{\left(i\right)}$, respectively, i=1,2,3 [17].

**Figure 2 materials-15-08906-f002:**
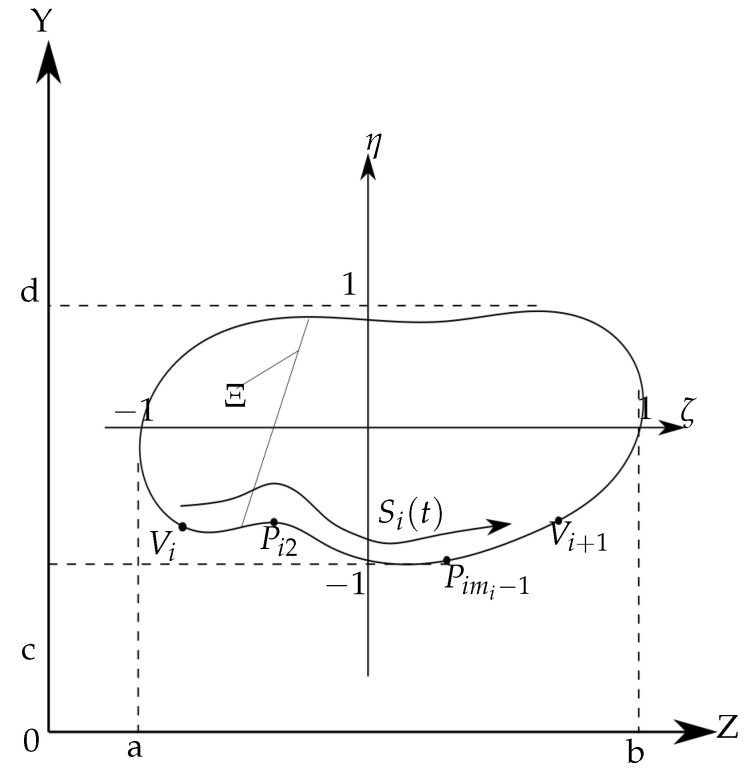
An arbitrary domain in initial and local coordinate systems.

**Figure 3 materials-15-08906-f003:**
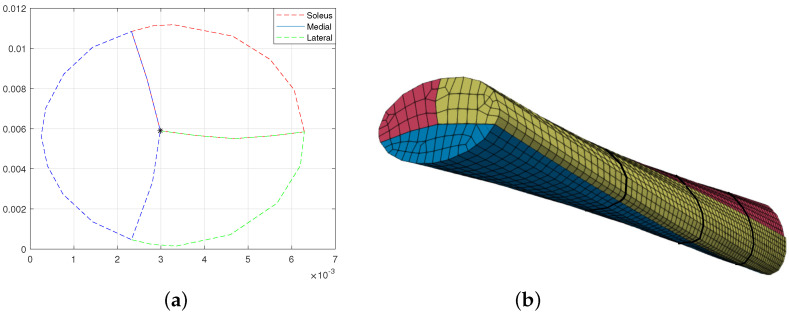
The Type III tendon representation. (**a**) The Achilles sub-tendons’ cross sections. (**b**) The pretwisted underformed Achilles tendon discretized by four ANCF-based continuum beam elements at each subtendon with ψ= 45°.

**Figure 4 materials-15-08906-f004:**
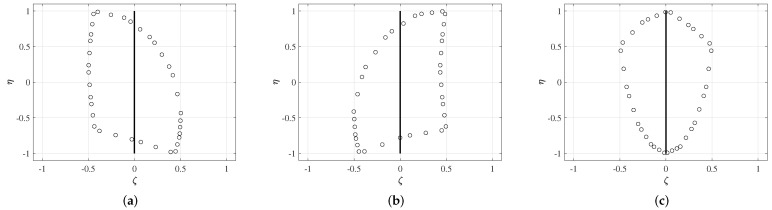
Integration approximations of the three subtendons by the Gauss–Green cubature formula. (**a**) Soleus. (**b**) Medial gastrocnemius. (**c**) Lateral gastrocnemius.

**Figure 5 materials-15-08906-f005:**
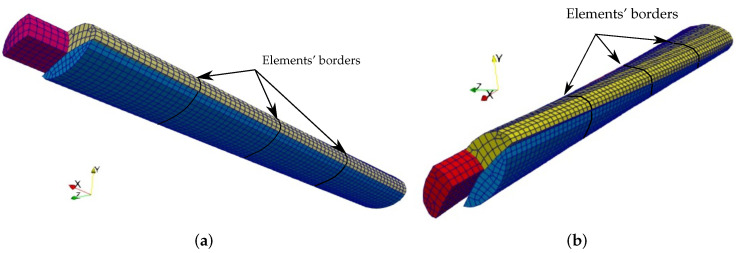
The deformed shapes of the pre-twisted Achilles tendons when the soleus is loaded and discretized by four ANCF-based continuum beam elements at each subtendon. (**a**) $\psi =15^{\circ }$ (**b**) $\psi =45^{\circ }$.

**Table 1 materials-15-08906-t001:** The elongation test results in [mm] for the straight and pretwisted soleus subtendon of Type III tendons from the neo-Hookean material model.

Elongation [mm] of the Soleus Sub-Tendon			
**Applied Load**	**Variation of $\psi^{\circ }$**		
**[N]**	ψ=0	ψ=15	ψ=45
10	0.143	0.146	0.168
20	0.286	0.289	0.313
30	0.430	0.433	0.457
40	0.574	0.578	0.602
45	0.647	0.650	0.675
60	0.864	0.868	0.893
80	1.157	1.160	1.186
90	1.304	1.307	1.333
100	1.451	1.454	1.481
150	2.197	2.201	2.228
200	2.958	2.961	2.989
300	4.524	4.527	4.557
400	6.151	6.155	6.187

**Table 2 materials-15-08906-t002:** Elongation results in [mm] for several mesh refinements for the straight and pretwisted soleus sub-tendon of Type III from the neo-Hookean material model under N=400 N tensile force.

Elongation [mm] of the Soleus Sub-Tendon			
**Element Number**	**Variation of ψ**		
**per Sub-Tendons**	$\psi =0^{\circ }$	$\psi =15^{\circ }$	$\psi =45^{\circ }$
$n_{\mathrm{Sol}}\times n_{\mathrm{MG}}\times n_{\mathrm{LG}}$			
1×1×1	6.051	6.055	6.123
2×2×2	6.089	6.093	6.123
4×4×4	6.151	6.155	6.187

## Data Availability

Not applicable.

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
