# Peer review of "Modeling of the Achilles Subtendons and Their Interactions in a Framework of the Absolute Nodal Coordinate Formulation"

_materials, 2022, doi:10.3390/ma15248906_

Round 1

Reviewer 1 Report

The paper entitled, `Modeling of the Achilles sub-tendons and their interactions in a framework of the absolute nodal coordinate formulation'.  The following issues need to be addressed:

Title

1. Suggestion of a title to be:

Finite element modeling of the Achilles sub-tendons and their interactions in a framework of the absolute nodal coordinate formulation(ANCF)

Abstract

2.  Abstract must be rewritten.  Follow this order starting with the introduction, issues, objectives, methodology, results, and conclusion.

Introduction

3.  Research objective of the study must be clearly stated in this section.

4.  State also other frameworks available besides the ANCF-based continuum. Justify why do you choose the ANCF framework.

5. In the abstract section, line 7, and also in line 52, the author keeps emphasizing decreasing computational costs. Can it s been justified?

ANCF beam element

6.  It is expected the author show anything unique about the model to fit the objective of the study.  Otherwise, the reader can always refer to the textbook for further reading to understand the equations used.

7.  The author needs to show clearly the formulation of the contact description at the beam axis as stated in the abstract line 11.  Which equation is proposed or derived by the author?  And how it is unique?

Numerical examples

8. Line 190, Cite `previous studies found.....', and give a citation for the work the author is referring to.

9.  Rearrange this section in this order: 

9.1 Show the model with dimensions.  Figure 3, and Figure 4(a-c) are taken from other journals or it is produced by the author?  same 

9.2 Show the model setup, assumptions made, loading conditions, mesh type, number of nodes, and number of elements. 

9.3  Show also, how the validation is made.

10.  Table 1, Table 2, and Figure 5 need more elaboration and explanation.

Conclusion

11. Rewrite the conclusion.  The conclusion should be short and concise.  The author needs to restate the research objective.

12.  Line 227, Avoid vague statements such as feasible.  Back the statement with data and results.

13.  How the model used can save computational time?   It is faster by how many a number of times?

References

14.  The references used are current and up to date.  20% of the paper are from year 2020 -onward.

Author Response

Reviewer Point P1.1 —             Suggestion of a title to be: Finite element modeling of the

Achilles sub-tendons and their interactions in a framework of the absolute nodal coordinate formulation(ANCF).

Reply: The authors thank the reviewer for the suggestion, however, in that case, there would be a tautology in the sense that absolute nodal coordinate formulation is an approach within Finite element modeling.

Reviewer Point P1.2 — Abstract must be rewritten. Follow this order starting with the introduction, issues, objectives, methodology, results, and conclusion.

Reply: That pattern is presented in the paper’s abstract as follows.

Introduction: The experimental results have revealed the sophisticated structure of the Achilles tendon (AT), including its material properties and complex geometry. The latter incorporates a twisted design and composite construction consisting of three sub-tendons. Each of them has a nonstandard cross section.

Issues: All these factors make the AT deformation analysis computationally demanding. Generally, 3D finite solid elements are used to develop models for AT because they can discretize almost any shape, providing reliable results. However, they also require dense discretization in all three dimensions, leading to a high computational cost.

Methodology: One way to reduce degrees of freedom is the utilization of finite beam elements, requiring only line discretization over the length of sub-tendons. However, using the material models known from continuum mechanics is challenging, because these elements do not usually have 3D elasticity in their descriptions. Furthermore, the contact description is formulated at the beam axis instead of a more general surface-to-surface description.

Results: This work studies the continuum beam elements based on the absolute nodal coordinate formulation (ANCF) for AT modeling. ANCF beam elements require discretization only in one

direction, making the model less computationally expensive.

Conclusion: Recent work demonstrates that these elements can describe various cross sections and materials models, thus allowing the approximation of AT complexity.

Reviewer Point P1.3 — Research objective of the study must be clearly stated in this section (introduction).

Reply: The research objective is given in the text as follows.

This research introduces a new approach to the deformation analysis of the Achilles sub-tendons and their interactions. The main idea is to use one-dimensional finite element discretization over the sub-tendon’s length (in the longitudinal direction) to decrease computational costs. It can be achieved by considering the tendon as a beam-like structure using ANCF-based continuum beam elements with specific descriptions for geometrically complicated deformable cross sections.

Reviewer Point P1.4 — State also other frameworks available besides the ANCF-based continuum. Justify why do you choose the ANCF framework.

Reply: Another only known way to create the Achilles tendon finite element model is to use 3D solid elements. That is presented in the manuscript as follows.

One of the most popular methods in modern biomechanical research for studying tendons is the finite element method (FEM). Using finite solid elements in a framework of the nonlinear FEM helps to comprehend the tendon’s complex geometry and materiality. However, AT modeling with solid finite elements leads to a significant number of degrees of freedom (DOFs) [11-13], which results in a long computation time to obtain a solution.

However, the reference list can be extended further, if it is necessary.

Reviewer Point P1.5 — In the abstract section, line 7, and also in line 52, the author keeps emphasizing decreasing computational costs. Can it s been justified?

Reply: In line 52, the authors want merely emphasize the necessity of the optimization for

AT models, such as they are quite computationally demanding, for example, in Knaus and Blemker (2021), the tendon model has over 16,000 elements. Regarding the justification of the possibility to make such optimization with the proposed ANCF-beam elements, the authors would like to refer to Obrezkov et al. (2020, 2021, 2022), where it was extensively demonstrated.

Reviewer Point P1.6 — It is expected the author show anything unique about the model to fit the objective of the study. Otherwise, the reader can always refer to the textbook for further reading to understand the equations used.

Reply: The authors agree with the reviewer, and novelty is necessary for a publication. The novelty of this manuscript is presented in the text as follows.

This research introduces a new approach to the deformation analysis of the Achilles sub-tendons and their interactions.

Thus, all the achievements are combined in one model, therefore, it is the first detailed model of the Achilles tendon based on beam elements.

Reviewer Point P1.7 — The author needs to show clearly the formulation of the contact description at the beam axis, as stated in the abstract line 11. Which equation is proposed or derived by the author? And how it is unique?

Reply: The contact description is presented in Eqs. (28, 29), which novelty is it is the contact formulation for deformable beams, defined at the beams’ centerlines and at the same time describes the surface-to-surface formulations. The assessment of the contact penetration is presented in Eq. (27) via the gap function. The presented formulas only need to describe the contact of three interconnected beams, and the authors are afraid that the further extension of the paper can even decrease the reader’s interest.

Reviewer Point P1.8 — Line 190, Cite ‘previous studies found.....’, and give a citation for the work the author is referring to. Rearrange this section in this order:

â—‹ Show the model with dimensions. Figure 3, and Figure 4(a-c) are taken from other journals, or it is produced by the author?

â—‹ Show the model setup, assumptions made, loading conditions, mesh type, number of nodes, and number of elements.

â—‹ Show also, how the validation is made.

â—‹ Table 1, Table 2, and Figure 5 need more elaboration and explanation.

Reply: The authors thank the reviewer for the comment and the manuscript is adjusted as follows.

â—‹ The references have been added.

â—‹ The dimension are given in the text as follows.

The length of the tendon was set at L = 0.07 m [29]. The geometrical results based on the approximation are 16.31 mm2 for the soleus sub-tendon, 15.98 mm2 for the medial and 19.57 mm2 for lateral sub-tendons, with total area equaling to 51.86 mm2. That slightly exceeds the average female tendon cross-section 51.2 mm2 and is smaller than the average male cross-section 62.1 mm2 [29].

Regarding the figures, they all are unique and drawn especially for this paper.

â—‹ The model setup and loading conditions are extended and given in the text as follows.

We used the Neo-Hookean material model with three shear modulus equal to c10 = 103.1 MPa for soleus, c10 = 143.2 MPa, and c10 = 226.7 MPa for medial and lateral sub-tendons, respectively [29]. We considered three different pre-twisted designs: ψ = 0, ψ = 15, and ψ = 45. The choice is based on the work [28], where the optimal value of twisted is found between 15 and 45 degrees. Then, the soleus sub-tendon was subjected to forces along the longest direction and applied at the last node, to the maximum applied tensile load of is 400 N. The applied force exceeds four times the loading conditions given in [15] and [28], allowing demonstrating non-linear deformations, about 10% of the initial length. On the other edge, r = 0 from (3) is fixed at the first node, such condition forbids the displacement but allows the cross-sectional contraction.

Regarding mesh type, in the work, the sub-tendon is described as beam-like structures and meshing only in a longitudinal direction. Regarding the number of nodes, and number of elements, the authors have considered different mesh refinements and demonstrated them in Table 2.

â—‹ Here we present an intermediate state of our research, where all previous achievements have been already verified with commercial finite element software and analytical solutions and are assembled into one model. The main idea is to show that not only by parts, but all system is workable, and AT tendon can be presented as beam-like structures at sub-tendon level from FE point of view.

â—‹ Table 1: The elongation test results in [mm] for the straight and pre-twisted soleus sub-tendon of Type III tendons from the Neo-Hookean material model.

Table 2: Table 2 presents the converge tests , where the elongation results for a number of mesh refinements for the straight and pre-twisted soleus sub-tendon of Type III from the

Neo-Hookean material model subjected to N = 400 N tensile force are given.

Figure 5: The deformed shapes for straight pre-twisted ψ = 15 the tendons are given in Fig. 5 , where the shapes are discretized by four ANCF-based continuum beam elements at each sub-tendon

Reviewer Point P1.9 — Rewrite the conclusion. The conclusion should be short and concise. The author needs to restate the research objective.

Reply: The authors regret that the conclusion looks vague to the reviewer. The main idea is given in the first sentence. This work applies the continuum-based ANCF beam element to

describe the deformation of the human Achilles tendon under tensile loading.

The procedure of the work in a general way is presented as follows.

This work applies the continuum-based ANCF beam element to describe the deformation of the human Achilles tendon under tensile loading.

Then, the detailed description of each sub-step is following.

In the study, AT is presented as a combination of three sub-structures, pre-twisted and sliding one around the others. The contact between them is described with the segment-to-segment algorithm. The Gauss-Green cubature integration formula captures the sophisticated crosssection form of each sub-tendon. The Neo-Hookean isotropic material model describes the pure elastic response.

The outcome of the present work is given as follows.

The results show that the model is feasible, but more careful verification is necessary. That can include models built with conventional 3D solid elements and the comparison with experimental data.

Such as the work is an intermediate step, and possesses certain limitations, the authors think, it is worth mentioning. Therefore, they are given in the last sentences.

Additionally, the work possesses certain limitations. For example, the cross-section area is considered the same along the longitudinal axis for all sub-tendons. That is a substantial simplification, but there is no available geometrical data to approximate variable cross-section area.

Reviewer Point P1.10 — Line 227, avoid vague statements such as feasible. Back the statement with data and results.

Reply: The authors agree, and substituted the questionable word ”feasible”.

The results show that the model is workable, but more careful verification is necessary.

Reviewer Point P1.11 — How the model used can save computational time? It is faster by how many a number of times?

Reply: The assumption regarding computational efficiency is based on Obrezkov et al. (2022), where only one sub-tendon is considered, and a comparison with FE ANSYS model is presented. In this work, we have considered three sub-tendons combined in one model and all of them are based on the same scheme, therefore, we believed the statement of the computational efficiency holds. Regarding time, we consider it as a dependent parameter, such as it depends on the programming realization, CPU capacity, and other parameters, so, the comparison in Obrezkov et al. (2022) is given in the number of used DOFs.

Reviewer Point P1.12 — The references used are current and up to date. 20% of the paper are from year 2020 -onward.

Reply: The reason for this is simple. Although the tendons are under intensive investigation for a couple of decades Ekiert et al. (2021), the non-uniform displacements just recently caught the eyes of the researchers. Moreover, the method used in the work has been started to adapt for the description of incompressible materials in Orzechowski and Fra¸czeks (2015) and continued only in Obrezkov et al. (2020). Many other things are quite novel here too, the cross-section approximation for deformable beams Obrezkov et al. (2022) and the contact approach for them Bozorgmehri et al. (2023). Therefore, most of the literature is current.

Reviewer 2 Report

Generally, the manuscript is well written.

Presented studies are interesting, which concerned the modelling of the Achilles sub-tendons and their interactions with using of the ANCF beam elements. Autors explored the given method and modelled the Achilles tendon consists of three sub-tendons with each having a complicated cross-section shape. They obtained the results for the straight and for three different pre-twisted designs: ψ = 0, ψ = 15, and ψ = 45, which was subjected to the maximum applied tensile load of 400 N. I can not discus results, beacuse more careful verification is necessary of them and the comparison with experimental data. But i have some comments to the boundary conditions.

How was applided load and support of model? Which (where) DOF were locked?

I also noticed a missing unit MPa at c10 in line 207.

I accept this manuscript after minor revision.

Author Response

Reviewer Point P2.1 — Generally, the manuscript is well written. Presented studies are interesting, which concerned the modelling of the Achilles sub-tendons and their interactions with using of the ANCF beam elements. Authors explored the given method and modelled the Achilles tendon consists of three sub-tendons with each having a complicated cross-section shape. They obtained the results for the straight and for three different pre-twisted designs: ψ = 0, ψ = 15, and ψ = 45, which was subjected to the maximum applied tensile load of 400 N. I can not discuss results, because more careful verification is necessary of them and the comparison with experimental data.

Reply: The authors are grateful for a positive response and hope that the presented corrections will allow the reviewer to accept the manuscript. Regarding the verification and comparison data, there is not much experimental and geometrical data available, which was, for example, a reason to resort to a graphical editor. Additionally, only several known models demonstrating nonuniform across cross-sectional deformation exist Handsfield et al. (2017, 2020); Yin et al. (2021); Ekiert et al. (2021); Knaus and Blemker (2021); Funaro et al. (2022). Moreover, many of them, excepting Handsfield et al. (2017); Yin et al. (2021), use anisotropic models during their numerical experiments. It is understandable but questionable from the mechanical point of view Annaidh et al. (2013). Currently, some experiments are underway at the University of Jyv¨askyl¨a and can be added and implemented later as a further step in doing research. In this work, the authors want to extend their previous studies and demonstrate another way of building the Achilles tendon models via beam structures.

Reviewer Point P2.2 — But I have some comments to the boundary conditions. How was applied load and support of model? Which (where) DOF were locked?

Reply: The authors thank the reviewer for the comment, and the description of the force application was added to the manuscript as follows.

Then, the soleus sub-tendon was subjected to forces along the longest direction and applied at the last node, to the maximum applied tensile load of is 400 N. The applied force exceeds four times the loading conditions given in Obrezkov et al. (2022) and Yin et al. (2021), allowing demonstrating non-linear deformations, about 10% of the initial length.On the other edge, only r = 0 from (3) is fixed at the first node, such condition forbids the displacements but allows the cross-sectional contraction.

Reviewer Point P2.3 — I also noticed a missing unit MPa at c10 in line 207.

Reply: The authors thank the reviewer for such notice, it was corrected in the manuscript.

We used the Neo-Hookean material model with three shear modulus equal to c10 = 103.1 MPa for soleus, c10 = 143.2 MPa, and c10 = 226.7 MPa for medial and lateral sub-tendons, respectively Yin et al. (2021).
